# Complete Situational Awareness through the Sensing Harmonization of Connected Vehicles and Smart Infrastructure

## Abstract

Multi-agent systems, inspired by biological collective intelligence, can deal with complex tasks that far exceed the capability of single agents, which is based upon the interactions between agents and the environment. As an instantiate of a multi-agent system, connected vehicles supported by smart infrastructure have been considered the next generation of road mobility and attract significant attention given its potential in terms of safety enhancement, fuel efficiency improvement, and environmental sustainability. As the core of connected vehicle technology, multi-agent perception is to achieve the complete situational awareness of the complicated environment and serve as the foundation for collective intelligence. However, the effectiveness of multi-agent perception has been compromised in real-world scenarios due to the multi-agent heterogeneous feature extraction methods and the high communication cost. To bring the connected vehicles into real roads by addressing these fundamental challenges, this paper presents Memory-Informed Multi-Agent Perception (MI-MAP), which can overcome the heterogeneity of feature extraction and leverages the shared memory in a computation-and-communication-light method for enhanced situational awareness. Drawing inspiration from human inference, our approach employs a memory-informed mechanism that uses an attention-driven memory module to capture multi-agent semantic interactions and motion dynamics from temporal data, thereby enhancing cooperative perception capabilities. Extensive experiments conducted on various benchmark tasks show the superior scalability of our approach, particularly in addressing the fundamental problems of the multi-agent perception, thereby establishing its potential as a practical solution for resilient AI systems.

## 1 Introduction

Recent advances in single-agent autonomous systems, such as deep learning-based autonomous navigation and reinforcement learning frameworks in robot controllers, have achieved significant milestones. However, their isolated intelligence and limited sensing capabilities hinder their ability to handle complex environmental dynamics and inter-agent interactions. In contrast, multi-agent autonomous systems, mirroring the seamless communication and adaptability observed in biological collective ecosystems, improve the single-agent system through information sharing and collaborative interactions, enabling comprehensive situational awareness and holistic decision-making.

One such domain is multi-agent perception, which aims to synergize the shared information Vaswani et al. (2017) to provide enhanced situational awareness by detecting the inclusive road users based on supervised learning process Meng et al. (2024).

Such cooperative perception algorithms usually consist of a feature extraction backbone, generating the abstract features for each connected agent, and a multi-agent feature fusion module, aggregating the multi-agent perception features to detect the objects. However, existing approaches simplify the cooperative reasoning process by prototyping feature extraction as a homogeneous process that utilizes the same feature extraction backbone and directly shares the abstract features for global reasoning. In complex and large-scale multi-agent perception scenarios, different types of agents usually adopt different feature extraction neural networks generating heterogeneous features and may use different fusion strategies, significantly impeding the multi-agent reasoning process and following downstream tasks. More critically, in complex multi-agent cooperation perception, directly communicating with other agents through the features extracted from the homogeneous backbone leads to fragile multi-agent systems. Such feature sharing method only supports single-frame

communication in real deployment and easily exceeds bandwidth limitations when memory information are shared among multiple agents. The limited memory sharing capability make cooperation and advanced reasoning ability infeasible in real-world. In other words, due to the high communication and computational cost, existing approaches are only suitable for handling homogeneous single-frame cooperative perception, which assumes consistent multi-source features, ignores the inter-agent temporal correlation, and fail to leverage the potential of the multi-agent memory for scenario understanding. Progress in multi-agent spatiotemporal perception systems—which fuse heterogeneous observations over time—will thus be pivotal to building the next generation of large-scale, energy-efficient, adaptive, and robust AI system.

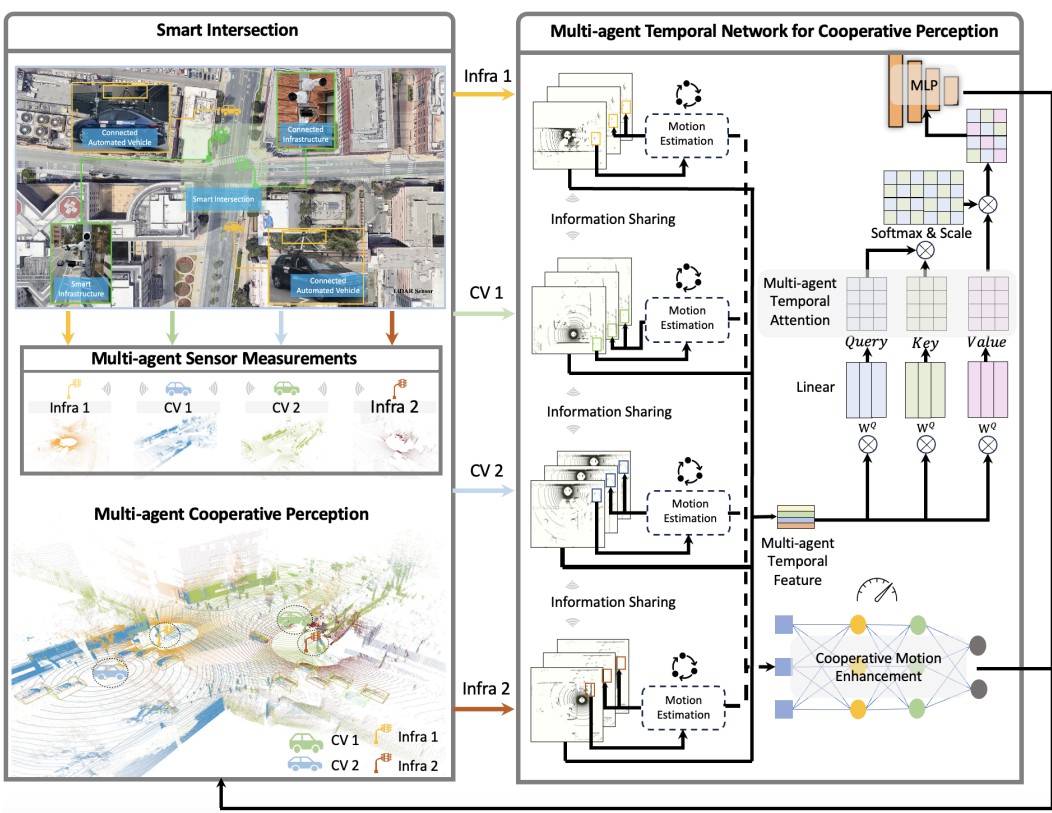

Figure 1: **Overview of the smart intersection and the conceptual Memory-Informed Multi-Agent Perception (MI-MAP) framework and architecture.** (a) Different connected agents share the collected information. (b) MI-MAP perception is designed to sample the multi-agent temporal information in each frame through an motion estimation method. A multi-agent temporal attention and motion enhancement network to fuse the selected temporal information from different agents.

Therefore, we aim to revolutionize existing multi-agent perception systems for deployment across multiple intelligent robot operating contexts by fully adopting a unique semantic and temporal attention fusion strategy and mimicking the human memory reasoning process through a unified communication-efficient procedure designed to accommodate heterogeneous types of agents in real-world scenarios. To suit for large-scale deployment of the multi-agent perception system, the cooperation strategy must address three critical problems: (1) Given the heterogeneous agents reasoning the situation through different deep neural network, the multi-agent feature fusion should establish a unified procedure to overcome the heterogeneity of features extracted by connected agents. Moreover, the shared agent information should be easily utilized by different agent's learned parameters of neural network to complete situational awareness. (2) Traditional multi-agent perception systems often involve every agent broadcasting entire abstract features, leading to redundant information sharing, communication overhead, network congestion, and latency of decision-making processes. However, in complex real-world environments, large multi-agent AI systems require communication-efficient methodologies to ensure seamless collaboration, making it essential for

each agent to dynamically share critical information while minimizing bandwidth usage. Therefore, dynamically selecting and transmitting only the most salient features from raw sensor data is crucial for enabling efficient and scalable multi-agent cooperation in AI systems.

Motivated by the challenge of real-world deployment of multi-agent systems and the benefit of using agents' memory for complete situational awareness through the harmonization of sensing capability from CVs and smart infrastructure, this work provides the Memory-Informed Multi-Agent Perception (MI-MAP) framework to address the fundamental challenges including the heterogeneity of agent feature extraction and limitation of communication in V2X cooperative perception. The framework of MI-MAP is illustrated in Fig. 1. We propose a multi-agent appearance and motion enhancement network to aggregate the object's temporal features across different CVs. By using the attention mechanism to process the appearance information from both temporal and spatial dimensions of each agent, the proposed multi-agent multi-frame appearance enhancement module explores spatial-temporal dependencies among multi-agent proposal features and fuses them into global representations for final bounding-box refinement. To effectively sample the historical appearance information for each connected agent, we design an motion estimation module to sample the historical information. The historical frames are chosen to be a small portion of the original point clouds instead of the learned feature to share among the different agents. We choose the original point clouds to overcome the heterogeneous issues because the learned representations extracted from the deep networks can become too domain-specific, making it difficult for them to be used by heterogeneous agents from different companies Xu et al. (2023) and generalized to temporal domains Kawaguchi et al. (2017). Therefore, the contribution can be summarized as below:

• The MI-MAP is the first-of-its-kind framework that balances flexibility, communication efficiency, and generality. MI-MAP can significantly reduce the volume of historical data from Multi-Agent. It can be generalized to various frameworks to reduce the size of the multi-agent shared information, substantially decreasing the data-sharing load and the computational and memory demands, while accommodating heterogeneous types of agents to tackle more complex and intricate tasks. Moreover, the MI-MAP also offers a highly adaptable solution for complete situational awareness of genetic applications, making it compatible with other multi-agent intelligence applications. The proposed method for complete situational awareness is verified in real-world environments, proving the superiority and potential for real-world dataset of our MI-MAP framework.

• We propose the method to sample the most valuable information within the region proposals from the various kinds of agents, instead of relying on agent extracted deep features, which could be more beneficial for the heterogeneous settings of the multi-agent systems and the enhanced situational awareness. Such a method could avoid the complicated multi-agent feature domain adaption, enhance the comprehensive reasoning process, and at the same time reduce the computational load.

• We introduce a memory-informed spatial–temporal fusion and motion-enhancement module that samples compact, raw point-cloud "memories" via constant-velocity motion estimation and fuses them with multi-agent appearance and temporal attention to encode cross-agent, cross-frame dependencies for robust 3D detection.

## 2 RELATED WORK

A central line of cooperative perception work treats inaccurate poses, time skew, and viewpoint changes as first-class citizens and aligns collaborator information before fusion. Yang et al. (2025) proposed lightweight, model-agnostic plugins—importance-guided query proposals coupled with deformable cross-attention—to correct feature-level misalignment in collaborative 3D detection, showing sizable gains under pose noise. Temporal asynchrony is addressed by estimating motion/feature flow in BEV and warping collaborator features to non-discrete delays (LRCP), or by trajectory-aware alignment that predicts feature flow along likely object paths (TraF-Align). Camera-centric V2I systems, proposed by Wang & Nordström (2025), learn frame-adaptive synchronization in BEVSync, further reducing cross-agent mis-timing.

Beyond one-shot fusion, recent methods, suggested by Wang & Nordström (2025), cache historical BEV features and learn spatio-temporal correspondences so that messages carry motion-aware updates rather than static features, improving resilience to motion blur and intermittent links. Representative designs include asynchronous flow predictors and attention modules that explicitly couple multi-time with multi-agent fusion.

With realistic bandwidth/latency budgets, Xu et al. (2025) designed what to send and how to fuse it. CoSDH formulates supply–demand-aware region selection and hybridizes intermediate- and late-fusion to preserve accuracy as bandwidth shrinks. Gan et al. (2025) proposed task-oriented/semantic

communication pipelines (e.g., SComCP) to learn end-to-end encoders that transmit only task-relevant semantics robust to channel impairments, while compact message units such as point clusters decouple sparse structure from high-level semantics to maximize information per bit. Complementary work, designed by Qiu et al. (2025), prunes background via map-aligned masks to further cut payloads. To improve interoperability across vendors and protect privacy, Fadili et al. (2025) proposed an late fusion method to aggregate only detections with principled uncertainty handling. Diffusion-based, designed by Huang et al. (2025), method has emerged as a means to hallucinate clean features and to jointly calibrate pose/time via learned generative priors (CoDiff), pointing to a promising direction for reconciling heterogeneous partners and noisy links. Tang et al. (2025) suggested to integrate multi-agent and multi-time aggregation in one stack using deformable attention to sample only informative regions across agents and frames, reducing cubic attention costs while preserving long-range context.

# 3 METHODS

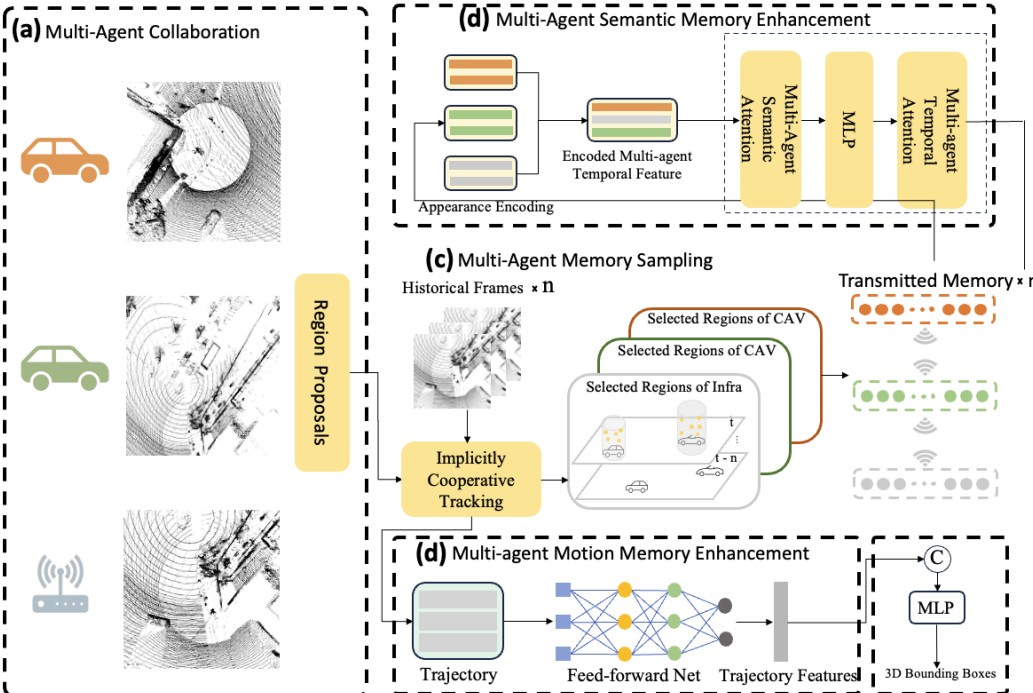

Figure 2: **Appearance and motion guided multi-agent 3D temporal detection.** MI-MAP consists of multi-agent collaboration, multi-agent region proposal generation, multi-agent multi-frame feature sampling, and the semantic and motion enhancement module. Initially, multi-agent region proposals are generated by the multi-agent fusion network, serving as inputs to the motion estimation module for efficient sampling of historical data. Subsequently, the sampled multi-agent temporal information is taken by the semantic and motion enhancement module to learn the spatial and temporal interactions to cooperatively detect the objects in 3D space.

**Overall Architecture.** As outlined in the introduction, when designing large-scale multi-agent perception under diverse and complex scenarios for complete situational awareness, the heterogeneity of agent-specific features and memory of the sensor data is crucial to achieve the harmonization of the CVs and smart infrastructure. The heterogeneity of multi-agent features impede the fusion process and the following scenario understanding. In addition, integrating memory into the cooperative perception algorithm markedly improves performance, particularly in challenging conditions such as occlusions, targets beyond sensor range, or sparse point clouds. Yet, leveraging multi-agent historical memory to enhance cooperative perception significantly increases the volume of the multi-agent shared data and computational burdens. To tackle the above problems, the proposed MI-MAP is specifically designed to seek a unified and efficient sharing procedure for multi-agent system to

seamlessly fuse the multi-source data and efficiently utilize multi-agent historical memory to ensure both streamlined data transmission and optimized computational load.

Fig. 2 presents an overview of multi-agent 3D temporal perception, where each agent collects the sensor measurements and shares the extracted features (e.g., detection outputs, raw sensory information, intermediate deep learning features) from multiple agents (e.g., CVs and smart infrastructure) are shared for complete situational awareness of the environment. Upon receiving the shared features from different agents, each CV can then generate region proposals to indicate the possible target in their surrounding environment. Subsequently, every CV samples local point cloud features from the LiDAR for those inclusive roadway objects, which are utilized by the motion estimation module to infer the possible historical regions of the objects. The predicted temporal regions from multiple agents will be processed by our proposed memory-informed multi-agent appearance and motion feature fusion network to enhance cooperative perception. Through such a manner, cooperative agents not only could fuse the memory information about nearby roadway objects across time and space dimensions in an efficient manner but also overcome the heterogeneity feature issues encountered in large and diverse multi-agent cooperation systems.

**Multi-Agent Region Proposal Generation.** MI-MAP adopts flexible approaches to generate region proposals. Each agent possesses the ability to generate region proposals independently, either through its own sensor measurements or by utilizing shared connected vehicle information. After region proposals are generated by individual agents using their respective methods, the proposed framework then integrates the multi-agent temporal data, enabling a robust and adaptable fusion of region proposals across agents. In the MI-MAP, we leverage either the intermediate features or preliminary bounding boxes to generate the region proposals, considering the rich information the intermediate features provide and the efficient transmission of bounding boxes in practice. Specifically, for the intermediate features, we utilize V2X-ViT Xu et al. (2022) to generate the multi-agent features and the region proposals due to its demonstrated effectiveness in multi-agent feature fusion and its relevance as a representative model in cooperative perception. V2X-ViT first uses the Point-Pillar Lang et al. (2019) backbone to extract features from the LiDAR point cloud in each agent and then utilizes the transformer to fuse the features from multiple agents. The fused features are utilized to generate the multi-agent region proposals. In naive late fusion, each CV will predict the bounding boxes with confidence scores independently and broadcast these outputs to the ego vehicle. Non-maximum suppression (NMS) will be applied to these proposals afterward to generate the region proposals.

**Multi-agent Multi-frame Feature Sampling**

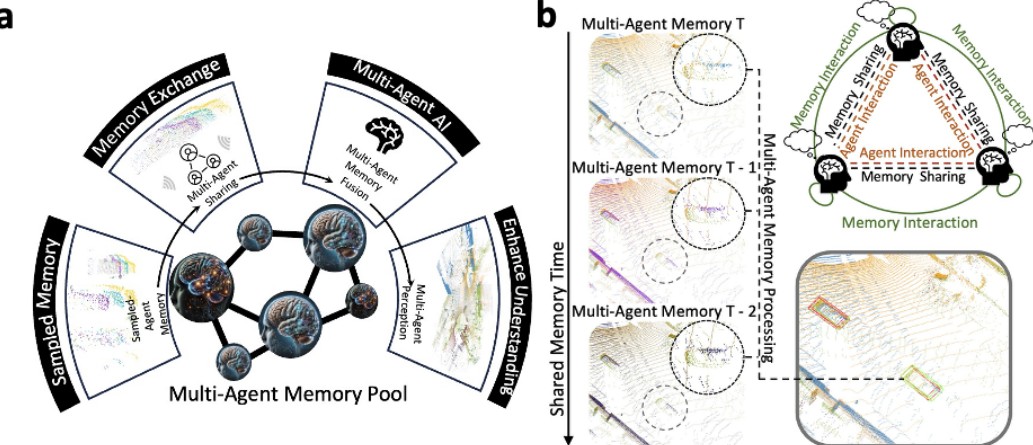

Figure 3: **The cooperative object detection performance under challenging scenarios.**(a) The overall memory reasoning process of multi-agent system. (b) The prior memory information of each agent is used to enhance the scenario understanding.

Fig. 3.a presents an overview of the sampling and reasoning process, which samples memory information from each agent to reduce the total volume of data. Then each agent exchanges and fuses the multi-agent memory information to achieve the comprehensive understanding of the surrounding environment. To aggregate multi-agent historical appearance and motion information to

enhance object detection performance, we propose to leverage the historical features in different memory timestamp, as shown in Fig. 3.b, to address the occlusion, sparsity, and out-of-range issues of LiDAR-based object detection. Specifically, after generating the multi-agent region proposals, the objective of multi-agent multi-frame feature sampling is to sample the points within the historical regions based on the proposed motion estimation module. The sampling approach focuses on the local point cloud data within the region proposals and samples the points from sequential frames. Our sampling method employs the constant velocity module, motivated by Weng et al. (2020), to predict and select local features in historical region proposals. The constant velocity model is designed to infer the historical positions of region proposals and subsequently select the relevant historical features within these regions. Unlike the explicit cooperative object tracking method, our motion estimation accelerates the training process by simplifying the 3D association, trajectory update, and management tasks. This strategy not only accelerates the training process and facilitates the retrieval of historical features but also avoids the complex association process typically involved in cooperative object tracking. We utilize the point cloud information inside the bounding boxes for two essential considerations: (1) Compared to the features that contain the bias due to the feature extraction backbone, the sampled point cloud information contains the raw representation of the surrounding objects. (2) Different agent's point cloud memory could be seamlessly shared with other agents and directly utilized by the multi-agent fusion module without further finetuned and complex designs.

Through this process, our method significantly reduces the memory volume and the computational load required to process multi-agent shared information. Consequently, our results, as visualized in Fig. 5 (d) and (e), attest to the practical deployability of proposed method in real-world settings, as it effectively improves the multi-agent detection accuracy and balances data volume and computational efficiency, ensuring its practicality for real-time applications in resource-constrained environments. After obtaining the predicted proposals, the connected agents could sample the local point cloud features within their respective sensor output. We adopt a uniform sampling method to pool the points from the non-empty voxels of the point clouds within the region proposals. Through such a sampling process, the semantic information for the surrounding objects, including their appearance, shape, and size information, could be maintained. Each connected agent only needs to transmit the sampled features instead of the overall point clouds and the shared point cloud can be used by any agent in the overall multi-agent system.

**Multi-Agent Multi-frame Appearance and Motion Enhancement** Although the historical multi-agent proposals and their captured points cloud object provide richer information to estimate the 3D bounding box more accurately, aggregating the multi-source information from a long sequence of object points has not been studied and remains challenging. Therefore, we propose the multi-agent multi-frame appearance and motion enhancement model to learn the historical appearance and motion interaction between different agents. Fig. 2 provides an overview of network design, where the multi-agent multi-frame appearance and motion enhancement module encodes the appearance and motion information and fuses the features to enhance the cooperative perception performance. The object geometry and motion features are separately encoded through appearance and multi-layer perceptron (MLP). Then, the encoded multi-agent multi-frame appearance features are processed by our proposed network to learn the appearance interaction between different agents across different times. The motion features of objects are extracted through MLP and concatenated with the appearance features to infer the 3D bounding boxes.

The first component of the multi-agent multi-frame appearance enhancement network, as shown in Fig. 2, aggregates the features of different agents and encodes the transmitted region. Specifically, for the appearance feature encoding, the multi-agent transmitted point cloud appearance regions will be aggregated together and then encoded through MLP to extend the dimension of the point cloud features. After obtaining the transmitted region, we first uniformly sampled $N$ points among the multi-source points. The sampled multi-agent points are used to calculate the relative offsets between each point and the nine key points (eight corner points plus one center point) of the region proposal. The resulting offsets are then converted to spherical coordinates and transformed, via MLP, to a geometric embedding of dimension $D$ that encodes the spatial correlation between the sampled point and the proposal box. The encoded features are then formed as the input to the multi-agent multi-frame appearance enhancement module. The above sampling and encoding approach aggregates the multi-agent information from the proposal points, which serve as the multi-agent representations of each frame to facilitate the multi-frame feature interaction.

After generating the multi-agent feature for each frame, MI-MAP employs the attention mechanism to explore spatial-temporal interaction among multi-source temporal features and process them into

global representations for final bounding-box refinement. Since the encoded multi-agent features contain the object features observed by different agents at different timestamps, we propose to utilize the self-attention mechanism Vaswani et al. (2017) to enhance the spatial-temporal relationship and point dependencies in the multi-agent proposals. We propose two attention operations, multi-agent appearance attention (MAA) and multi-agent temporal attention(MAF). MAA is utilized to calculate the semantic affinities between different points within a sequence of multi-source data. After extracting the semantic features, MAF first fuses the global temporal features of the semantic features and then calculates the temporal affinities among the multi-agent temporal features.

Specifically, MAF leverages the encoded multi-agent temporal feature to focus on the appearance information among different agents. The selected temporal regions $F_i \in \mathbb{R}^{N_i \times (P \times T) \times D}$ from the agent $i$ will be encoded into the three-dimensional matrix, where $N_i$ represents the multi-agent points sampled from the agent $i^{th}$, $P$ and $T$ represent the number of region proposals and the number of frames, and $D$ is the dimension of each point. Then, we aggregate the multi-agent points as $F \in \mathbb{R}^{N \times (P \times T) \times D}$ to conduct self-attention to learn the spatial interaction between the multi-agent features as:

$$M_a = MultiHeadAttn(Q(F), K(F), V(F)) \tag{1}$$

The $Q(F), K(F), V(F)$ represent the learned matrix as query, key, and value. $MultiHeadAttn$ is the multi-head attention mechanism. The attention feature $M_a$ is then processed by MLP and MAF to calculate the temporal affinities among the multi-agent feature $M_a$.

After learning the multi-agent interaction, our MAF attention aims to propagate information of the multi-agent points across different frames to capture richer whole-sequence information. MAF aggregates temporal cues across successive timestamps to capture their underlying relationships. Specifically, we encode the temporal information by using $mean$ and $max$ operations to $M_a$ and learn a unified representation by using the MLP, which could be formulated as follows:

$$M_a^{'} = Sigmoid(MLP(max(M_a), mean(M_a))) \tag{2}$$

The generated feature $M_a^{'}$ encodes the high-level representation of $M_a$ for each frame and is concatenated with $M_a$ shifted forward by one time step. Finally, an MLP then learns the temporal correlations and outputs the resulting spatio-temporal feature $M$.

In addition to spatial-temporal features $M$ extracted from the multi-agent point clouds, we also exploit PointNet Qi et al. (2017) to extract the motion information of the objects. The objects' trajectories obtained from the motion estimation module are used as the motion information of the FFN. The network takes the multi-agent trajectory as input to extract the embedding of motion information and provides the movement information of the surrounding objects to enhance the cooperative perception. The extracted motion features are then concatenated with the appearance features in the detection head.

## 4 EXPERIMENTS

**Dataset** To evaluate our approach, we have conducted experiments on the real-world large-scale cooperative perception dataset V2X-RealXiang et al. (2024). V2X-Real is a real-world dataset with diverse sensor data from CVs and smart infrastructure collected at the public intersection. The camera, radar, LiDAR, global navigation satellite system (GNSS), and inertial measurement unit (IMU) have been adopted to capture the interactions between diverse road users in urban environments with diverse traffic. Scenarios where objects are far away from the ego vehicle, and occlusion, sparsity, and out-of-range issues happen frequently and challenge situational awaress is included in the dataset. The sensors are mounted on top of each CV and smart infrastructure. Ground-truth data for inclusive roadway objects with 3D bounding boxes and their unique ID are provided. More details of dataset can be found in Xiang et al. (2024).

**Performance** In Fig. 4, the performance of MI-MAP and V2X-ViT on the V2XSet and V2XReal dataset is shown. With selected temporal information shared from each agent, our method improves the overall AP@0.7 by 4.9% and 7.1% separately. As discussed previously, the significant improvement is due to our proposed method MI-MAP which effectively exploits the temporal features to enhance the cooperative object detection performance except for leveraging the shared features from multi-agent. The multi-agent temporal features processed by our proposed MI-MAP appearance and motion enhancement module contribute to the accuracy improvement of cooperative perception and accordingly situational awareness. Furthermore, effectiveness in real-world scenarios demonstrates

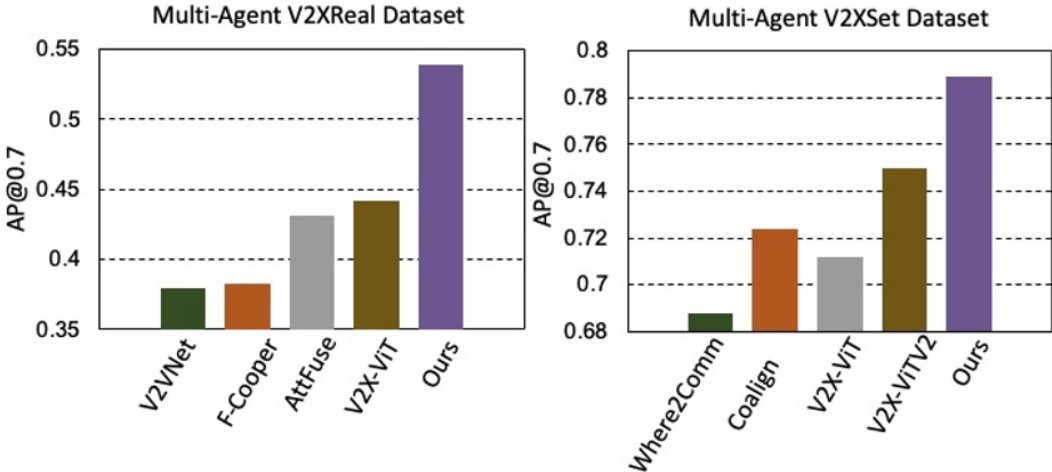

Figure 4: **Overall cooperative perception performance.** The Average Precision (AP) on the multi-agent V2XReal and V2XSet datasets was achieved by existing multi-agent perception methods.

that our design, leveraging a small portion of the temporal information, can significantly improve the cooperative perception and achieve robust perception performance.

**Volume of shared data** The proposed MI-MAP has been verified in the real-world V2X-Real dataset to address the critical challenges when applying the multi-agent perception algorithm in the real world. (1) In the MI-MAP, the total volume of the temporal data collected by multi-agent system has been reduced significantly. (2) MI-MAP supports the heterogeneous types of agent using different feature extraction backbone to accomplish the multi-agent perception tasks by seamlessly sharing both the bounding boxes and semantic information. We investigated the data volume of the multi-agent temporal information sampled by our method by calculating the number of transmitted points, as depicted in Fig. 5(a). The average and sampled multi-agent points per frame in both the real-world are calculated. Specifically, we sum the total number of points captured by all agents across all frames and then calculate the average number of points per frame. For our sampled multi-agent points, we multiply the number of region proposals for all agents and the sampled points within each proposal and average the total transmitted points for each frame. As Fig. 5(a) illustrates, MI-MAP reduces the per-frame multi-agent point-cloud payload by 89 % on V2X-Real and 92 % on V2X-Set, dramatically trimming the volume of data that connected agents must capture and transmit. This reduction is pivotal for the real-world V2X deployment, where bandwidth, latency, and compute budgets are tight. By shrinking the data payload per frame, MI-MAP lowers network congestion and end-to-end latency, enabling faster and more reliable inter-agent communication. The lighter payload also reduces the processing burden of fusing data from multiple agents, so the system can accommodate larger fleets without exhausting GPU memory or other computational resources, making the solution more scalable for larger networks and more complex urban environments.

**Attention Module Analysis** In addition, we further verify the ability of MI-MAP to fuse the multi-agent point cloud memory information and the proposed multi-agent temporal attention. We evaluate the effectiveness of the attention mechanism depicted in Fig. 1. We denote our multi-agent appearance and temporal attention as app and temp attention mechanisms for simplicity in Fig. 5(b). By comparing the histogram of Fig. 5(b), we observe that the performance drops 0.7 in AP@0.7 after removing the multi-agent temporal attention module. The dropped performance comes from the disappearance of temporal interaction among the different agents due to the multi-agent temporal attention module. It demonstrates that the interactions among different frames are essential for achieving accurate detection since the model can better integrate multi-view information from the whole 3D trajectory. To further demonstrate the effectiveness of the multi-agent temporal interaction, we explore the impact of incorporating historical frames on cooperative object detection performance.

**Motion Module Analysis** Furthermore, we investigate the impact of the multi-agent multi-frame motion enhancement module. We directly remove the multi-agent multi-frame motion enhancement component and at the same time other components in Fig. 1 remain the same. The performance of cooperative object detection, with and without the motion enhancement module, is illustrated in Fig. 5(c). We can observe that with motion information, the performance of cooperative object de-

tection improves by 1.7 % and 1.6 % on the simulation and real-world V2X dataset. The observed performance improvement indicates that historical motion information offers valuable temporal insights for accurate cooperative object inference, and our method of motion information extraction effectively integrates with appearance features, thereby enhancing cooperative object detection.

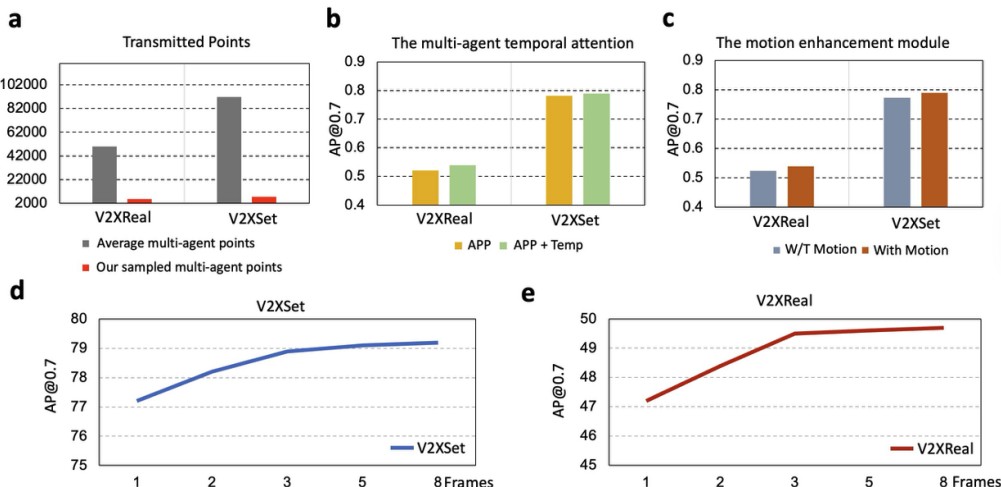

Figure 5: **The decomposed study of each module in the proposed method.** (a)(b) The effectiveness analysis of the sampled historical frame in our proposed cooperative perception framework. (c) The transmitted points of our proposed method. (d) The effectiveness analysis of the motion enhancement module. (e) The effectiveness analysis of our attention module.

According to Fig. 5(d) and (e), incorporating an increasing number of historical frames consistently enhances cooperative perception performance. This confirms that our proposed attention mechanism effectively captures the temporal interactions as short memory among multi-agent temporal points, thereby improving detection performance. Notably, upon integrating eight frames of historical information from various agents, we observe that performance gains plateau, indicating the saturation of the amount of historical data that contributes to performance enhancement. In our default setting, we only incorporate the three historical frames because it is important to balance the length of memory and the volume of data shared between agents. While incorporating additional multi-agent memory information enhances detection performance, it also increases the amount of data exchanged, which can lead to communication overhead and latency. Thus, careful consideration must be given to the trade-off between the benefits of including more temporal information and the practical constraints of data transmission in real-time systems. Maintaining this balance is essential for maximizing both detection accuracy and overall system efficiency.

## 5 CONCLUSION

In this work, we introduce MI-MAP, a framework that integrates sensing information through collective intelligence to improve situational awareness. Our enhance the multi-agent perception by sharing lightweight point-cloud memories among heterogeneous vehicles and infrastructure agents, furnishing a backbone-agnostic common space for bandwidth-efficient cooperative perception. By communicating raw, geometry-faithful memories rather than heavy intermediate features, MI-MAP suppresses both communication and computation costs. Our experiments in Fig. 5(c) reveal that the per-frame payload of transmitted points is reduced by 89 % on V2X-Real and 92 % on V2X-Set, greatly easing bandwidth and GPU demand in real-time deployments. Extensive experiments on the real-world V2X-Real benchmark demonstrate that MI-MAP lifts class-average AP@0.7 by 7.1% over the SOTA while simultaneously reducing the historical shared point clouds by 87 % of the overall raw points. More importantly, MI-MAP exchanges raw point-cloud memory that constitutes a backbone-agnostic sharing space, thus resolving feature heterogeneity issues and allowing every agent to synthesize its neighbors' historical views into a richer spatiotemporal context, advancing cooperative perception beyond the limits of single frame fusion.

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

# A APPENDIX

## A.1 EVALUATION ENVIRONMENT

To evaluate our approach, we have conducted experiments on the real-world large-scale cooperative perception dataset V2X-RealXiang et al. (2024). V2X-Real is a real-world dataset with diverse sensor data from CVs and smart infrastructure collected at the public intersection. The camera, radar, LiDAR, global navigation satellite system (GNSS), and inertial measurement unit (IMU) have been adopted to capture the interactions between diverse road users in urban environments with diverse traffic. The intersection is located at a busy intersection on the University of California, Los Angeles (UCLA) campus, with the sensors capturing very diverse road users, including VRUs, scooters, wheelchairs, cyclists, emergency vehicles, buses, trucks, and standard vehicles. Scenarios where objects are far away from the ego vehicle, and occlusion, sparsity, and out-of-range issues happen frequently and challenge situational awaress is included in the dataset. The sensors are mounted on top of each CV and smart infrastructure. Ground-truth data for inclusive roadway objects with 3D bounding boxes and their unique ID are provided. More details of dataset can be found in Xiang et al. (2024).

## A.2 SAMPLE DATA VISUALIZATION

The effectiveness of using this information can be seen in both Fig. 6, where the usage of the sampled multi-agent point cloud memory can significantly enhance the perception performance under challenging detection scenarios. And such process can avoid heterogeneous feature issues and ensure that the various agents can collaboratively use the information to enhance the scenario understanding.

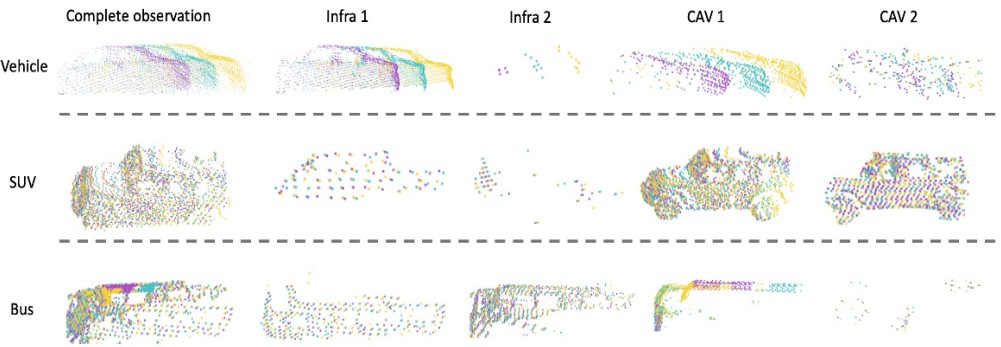

Figure 6: **The decomposed study of each module in the proposed method.** We visualize the temporal observations and shared information across connected agents. The temporal information is shown by different colors to emphasize its crucial role in constructing the object's point clouds. The moving vehicle, along with the sparsely observed sport utility vehicle (SUV) and bus, are supplemented by dense point clouds collected at various timestamps, facilitating cooperative perception accuracy.

## A.3 KINEMATIC MODEL

We employ the constant velocity kinematic model to predict potential positions for each proposal in a sequence of frames.
Given the region proposals $D(t) = \{(x,y) \mid x,y \in \mathbb{R}\}$ and the center $(x,y)$, the constant velocity model can be expressed as follows:

$$x_j^{'} = x_j + v_x \tag{3}$$

$$y_j^{'} = y_j + v_y \tag{4}$$

, where the subscript $j$ indicates the index of the region proposal and $v_x, v_y$ represents the different velocities used to update the region proposals.

## A.4 THE GENERALIZATION OF OUR METHOD TO LATE FUSION FRAMEWORK

In classical late-fusion pipelines Xu et al. (2021), each agent transmits its locally detected 3-D bounding boxes, and a central module performs geometric alignment and redundancy suppression.

To assess the portability of our approach, we augment that pipeline by attaching the MI-MAP sampled memory, which is approximately ten percent of the raw point cloud selected by the region proposals. Because the selection is performed directly in LiDAR space, the resulting subset is inherently backbone-agnostic and therefore compatible with detectors of arbitrary architecture. Crucially, Fig. 7 shows that incorporating a small number of points can significantly improve the performance of the model. Our method utilizing contextual points increases late-fusion performance by 7.7 % on V2X-Set and 10.6 % on V2X-Real.

**Late Fusion Framework Performance**

Figure 7: The performance comparison in the late fusion framework.

The improvement comes from the sampled memory that provides fine-grained geometric context. This geometric context corroborates or refines the coarse hypotheses conveyed by bounding boxes, thereby suppressing false positives and recovering occluded objects—benefits that are particularly pronounced on the more viewpoint-diverse V2X-Real data. At the same time, the compactness of the sampler ensures that accuracy gains do not jeopardize real-time operation. These results demonstrate that the memory-centric paradigm underlying MI-MAP transfers cleanly from intermediate-fusion to late-fusion settings, offering a drop-in, resource-aware enhancement that delivers double-digit improvements in detection precision without breaching practical V2X bandwidth or compute budgets.

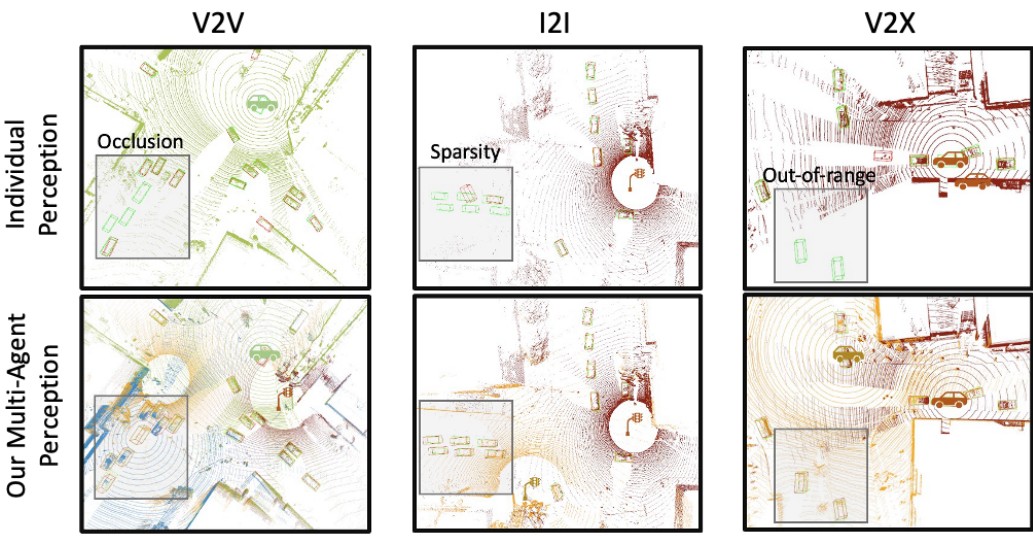

Figure 8: **Overall cooperative perception performance.** From left to right, we select different settings including the vehicle-to-vehicle, infrastructure-to-infrastructure, and vehicle-to-everything cooperation. Our multi-agent object detection framework successfully detects occluded, sparse, and out-of-range objects in various scenarios and settings.

## A.5 VISUALIZATION OF MODEL PERFORMANCE

The performance of MI-MAP is evaluated in the various scenarios of the V2X-Real dataset. Fig. 8 compares MI-MAP with traditional single-agent perception through different settings of agents, including the V2V, I2I (infrastructure-to-infrastructure), and V2X. V2V, I2I, and V2X mean that there is harmonization between CVs, between different infrastructure agents, and between CVs and infrastructure, respectively. Note that the infrastructure can provide a top view of the intersection through the LiDARs, which will complement the CV in terms of additional situational awareness. As can be seen in Fig. 8, even in challenging scenarios where objects are occluded, sparsely represented, and distant from the CV, the MI-MAP framework can still perceive objects accurately. The predicted red bounding boxes match the ground truth green bounding boxes, meaning the MI-MAP is able to handle these challenging scenarios. This is because, in our MI-MAP, the sensor data from multi-agent providing additional views from different angles is integrated by our proposed MI-MAP even if there are occlusion or point cloud sparsity issues for a specific agent. When no additional information is available and objects are occluded or far from the CV, the single-vehicle perception algorithm struggles to detect surrounding objects, as illustrated in Figure Fig. 8. As can be seen, there are no point clouds or sparse point clouds in the areas marked by the red rectangle, the network has no clues to detect the objects. However, in Fig. 8, those areas are covered by additional sensors mounted on the cooperative agents. Fig. 9 further compares the model performance under the challenging perception scenarios.

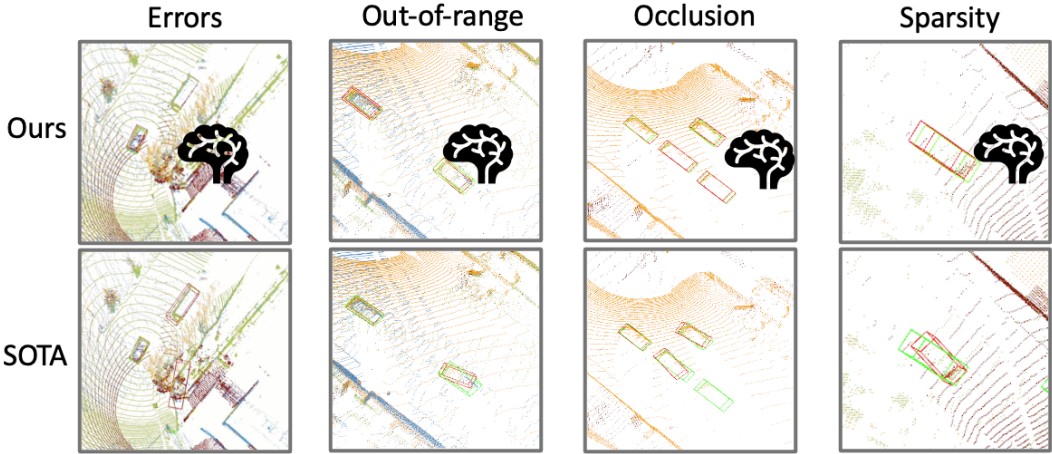

Figure 9: **The cooperative object detection performance under challenging scenarios.** We compare the errors, out-of-range, occlusion, and sparsity scenarios to illustrate our improvements. The first row gives the detection results of our MI-MAP. The second row is the results from state-of-the-art (SOTA) work V2X-ViT Xu et al. (2022). The green bounding and red boxes represent the ground truth and prediction of the objects.

