# OpenReview forum: "Complete Situational Awareness through the Sensing Harmonization of Connected Vehicles and Smart Infrastructure"
_ICLR.cc/2026/Conference — ICLR 2026 Conference Withdrawn Submission_

### Official Review · Reviewer_exmC · 2025-10-31

**Soundness:** 1
**Presentation:** 1
**Contribution:** 1
**Rating:** 0
**Confidence:** 4

**Summary:**

The paper proposes a multi-agent perception model enhanced with temporal attention modules to improve perception performance.

**Strengths:**

Fancy diagrams, I guess?

**Weaknesses:**

### Major Concerns
1. **Potential Misuse of diagram**. Figure 5(b) and 5(c) looks *identical*. I am not sure if the authors misuse the same diagram here. If not, this can suggest that the temporal attention and enhancement modules play an identical role in Average Precision, which is highly unlikely. Otherwise, it might suggest that the two modules have marginal effects, and the performance gap is only caused by noise.
2. **Misleading captions**. I am not sure how 5(d) and 5(e) respectively reflect the effectiveness of the motion enhancement module and the attention module. Based on the description in the main text, they should both reflect the effect of incorporating an increasing number of historical frames.
3. **No significant contribution**. Given the concerns above, it is reasonable to suspect what actually contributes to the performance gain in Figure 4. Suppose temporal attention and motion enhancement modules played the exact role in the final performance. Would it be fair to say the only factor that contributes to the final performance is the additional one or three frames from the past?

### Minor Issues
1. Duplicated symbol "(d)" in Figure 2.
2. The organization can be further improved. Currently, the paper appears to be a direct copy-paste from another template, such as a TRB AM paper.

**Questions:**

The only question I have for the authors is

**Why did you submit this unfinished paper that is apparently a direct copy from an existing short report without careful experiment design and sufficient results to demonstrate your contribution?**

**Details Of Ethics Concerns:**

NA.

---

### Official Review · Reviewer_R2P2 · 2025-10-31

**Soundness:** 2
**Presentation:** 2
**Contribution:** 3
**Rating:** 4
**Confidence:** 4

**Summary:**

This study proposes MI-MAP (Memory-Informed Multi-Agent Perception), a novel framework designed to achieve complete situational awareness through cooperative perception between vehicles and infrastructure. Conventional cooperative perception systems often suffer from heterogeneity caused by different backbone architectures across agents and high communication costs resulting from the sharing of raw data or features. MI-MAP addresses these limitations through memory sharing, sampling, and region proposal mechanisms. By sampling and sharing point clouds, the framework effectively mitigates heterogeneity while reducing communication overhead. Furthermore, by leveraging historical data and estimated motion information alongside a tailored fusion model, MI-MAP demonstrates superior perception performance.

**Strengths:**

- The study effectively addresses the challenges of heterogeneity and communication cost in V2X cooperative perception by leveraging raw point clouds and historical data.

- Instead of relying on commonly used features or bounding boxes, the authors employ raw data to innovatively overcome limitations inherent in intermediate fusion approaches.

- Rather than focusing on a single issue, the framework simultaneously tackles multiple challenges in cooperative perception, making it more suitable for real-world driving environments.

- The proposed method integrates previously fragmented components—region proposals, historical data, sampling, and attention modules—into a cohesive and effective framework.

- This work contributes meaningfully to the relatively underexplored design space between early and intermediate fusion in cooperative perception.

**Weaknesses:**

- The methodologies employed in MI-MAP build upon existing approaches such as region proposals, data sampling, and motion estimation, which have been widely explored in previous intermediate fusion studies on collaborative perception.

- The evaluation lacks sufficient comparison with prior works addressing heterogeneity. While performance is compared against several cooperative perception baselines, it does not include studies that specifically focus on heterogeneity mitigation.

- Comparative analysis with communication cost–oriented research is also limited. Although the paper discusses the volume of shared raw data, it does not adequately benchmark its communication efficiency against intermediate fusion approaches that prioritize communication reduction.

- Despite claiming to address heterogeneity, the proposed method does not fully resolve sensor-level heterogeneity, particularly between LiDAR and camera modalities. Since the framework assumes a LiDAR-based setting, it only partially tackles the broader heterogeneity problem in cooperative perception.

- The performance evaluation primarily relies on the AP@0.7 metric and baseline models that may not fully reflect the latest state-of-the-art. Incorporating comparisons with more recent cooperative perception approaches could further substantiate the validity of the experimental results.

**Questions:**

- Could the authors provide results comparing perception performance based on AP@0.3 and AP@0.5 metrics?

- Are there any comparative analyses of the volume of shared data between MI-MAP and late fusion or prior communication-efficient intermediate fusion methods?

- Is there a specific reason for focusing the performance comparison primarily on V2X-ViT, rather than including other recent cooperative perception research?

---

### Official Review · Reviewer_GN6K · 2025-11-04

**Soundness:** 2
**Presentation:** 1
**Contribution:** 2
**Rating:** 2
**Confidence:** 4

**Summary:**

This paper presents a cooperative perception method, operating under an early-fusion paradigm, designed to tackle the critical challenges of feature heterogeneity and communication efficiency. A key aspect of the proposed approach is its integration of temporal information from each individual agent.

Specifically, to address feature heterogeneity, the method selectively extracts crucial information from each agent's point cloud within identified region proposals. Subsequently, the authors introduce a spatial-temporal fusion and motion-enhancement module that is informed by memory. This module generates compact memories from raw point clouds by employing constant-velocity motion estimation. These memories are then fused with appearance features from multiple agents, utilizing temporal attention mechanisms, to capture dependencies across different agents and time frames for more robust 3D detection.
The authors validate their approach on the V2XReal and V2XSet datasets, reporting improvements in perception accuracy alongside a reduction in communication overhead.

**Strengths:**

1. This paper addresses two significant and relevant problems in the field of cooperative perception: feature heterogeneity and communication efficiency.
2. The figures in the paper are generally clear and effectively illustrate the overall workflow of the proposed method. The inclusion of point cloud visualizations is helpful for understanding the approach.

**Weaknesses:**

1. The proposed method appears to be a composition of existing techniques. This paper primarily combines established methods such as self-attention for point cloud sampling, motion enhancement, and feature fusion. It is unclear what novel, targeted modifications or designs have been introduced specifically for the cooperative perception scenario. Consequently, the technical novelty of the work feels limited.
2. The introduction and related work sections lack a comprehensive comparison with the vast body of recent literature in multi-frame cooperative perception, communication efficiency, and heterogeneous collaboration (covering early, intermediate, and late fusion). The authors fail to position their work adequately against existing methods, making it difficult to assess the paper's specific contributions and novelty.
3. The methodology section is difficult to follow. It is often unclear whether the authors are describing their specific algorithm design or simply reviewing common techniques. The paper lacks sufficient explanation of the design choices, and crucial details are absent from the equations, figures, and accompanying text, which makes it challenging to fully grasp the technical implementation of the method. The authors' narrative style needs significant improvement for clarity.
4. The experimental section suffers from several weaknesses:

      4.1 The set of compared methods is too limited. The absence of comparisons with state-of-the-art methods from the last two years makes it difficult to convincingly evaluate the superiority of the proposed approach.

      4.2 The comparison of communication volume is unconvincing. The authors only compare against a baseline that shares the entire raw point cloud, without including other efficient communication methods as benchmarks. This fails to substantially demonstrate the advantage in communication efficiency.

      4.3 The evaluation does not consider the impact of real-world collaborative noise, such as localization errors and transmission delays, on the model's performance.
5. This paper requires further revision for clarity and precision. For instance, the three key issues described on line 100 do not seem to align with the subsequent discussion. The related work section is missing relevant citations for some claims. Furthermore, many sentences are overly long and convoluted (e.g., lines 95-100), which hinders readability.

**Questions:**

See Weaknesses

---

### Official Review · Reviewer_4Y3A · 2025-11-06

**Soundness:** 3
**Presentation:** 3
**Contribution:** 3
**Rating:** 6
**Confidence:** 3

**Summary:**

The paper proposes MI-MAP, a cooperative perception framework for connected vehicles and smart infrastructure that shares compact, raw point-cloud “memories” and fuses them with attention over time and across agents. The method targets two practical issues—heterogeneous feature backbones and high communication cost—by avoiding backbone-specific feature sharing and using motion-guided sampling plus spatial–temporal attention to refine 3D detection. Experiments on V2X datasets indicate improved detection accuracy with substantially lower data transmission.

**Strengths:**

* The paper clearly identifies the real-world bottlenecks of cooperative perception—feature heterogeneity and communication bandwidth—and proposes a unified solution that effectively mitigates both.
* The introduction of a memory-informed spatial–temporal fusion strategy is novel, biologically inspired, and practically relevant for dynamic multi-agent environments.
* The use of raw point-cloud memory sharing instead of backbone-specific features is elegant, allowing interoperability among heterogeneous agents.

**Weaknesses:**

* The paper provides limited theoretical analysis of why the attention-based memory mechanism is stable and robust across heterogeneous agents.
* The robustness to time/pose noise, bandwidth jitter, and sensor asynchrony is not thoroughly evaluated.
* The writing is occasionally verbose and key implementation details (e.g., latency, GPU utilization) are under-reported for real-time claims.
* Comparisons to the most recent cooperative-perception baselines (e.g., deformable cross-attention or diffusion-based methods) are not comprehensive.
* The “human memory” analogy is lightly motivated and not tied to a formal cognitive model, which weakens the

**Questions:**

* How does the system handle temporal misalignment when agents have different frame rates or unsynchronized clocks?
* How sensitive are results to the number of historical frames, and where does performance saturate under stricter bandwidth limits?
* How does MI-MAP compare against learned communication/compression policies or information-bottleneck baselines at equal payload?
* What are end-to-end latency and throughput on realistic edge hardware, and how do they scale with the number of agents?
* How are conflicting or inconsistent observations across agents reconciled when building the shared memory for the same object?

---

### Official Review · Reviewer_A7mg · 2025-11-08

**Soundness:** 3
**Presentation:** 3
**Contribution:** 2
**Rating:** 4
**Confidence:** 3

**Summary:**

This paper proposes the Memory-Informed Multi-Agent Perception (MI-MAP) framework to address critical challenges in collaborative perception. Real-world deployment of collaborative perception remains limited by two key issues: heterogeneous feature extraction, since different agents often employ distinct neural networks that hinder feature fusion, and excessive communication overhead, as traditional methods transmit large high-dimensional features that exceed available bandwidth. MI-MAP introduces three core components to tackle these challenges. First, each agent generates region proposals using either local sensor data or shared information. Second, it samples raw point clouds from historical frames instead of abstract features, which mitigates heterogeneity and reduces data transmission. Third, it integrates Multi-Agent Attention (MAA) and Multi-Agent Fusion (MAF) modules with PointNet to effectively combine spatio-temporal and motion cues for more accurate 3D perception. Comprehensive experiments demonstrate that MI-MAP achieves notable improvements in perception performance, particularly under occlusion and sparse point cloud conditions, while significantly reducing communication costs.

**Strengths:**

1. A collaborative perception framework is proposed that balances flexibility, communication efficiency, and generality, enabling large-scale deployment of heterogeneous agents. The motivation is interesting and useful.

2. A region-proposal-based raw point cloud sampling method is designed to avoid feature heterogeneity while reducing both computational and communication costs.

3. The spatiotemporal attention and motion-enhancement module leverages historical memory to improve perception robustness in complex scenarios, achieving outstanding performance on the V2X-Set and V2X-Real datasets.

**Weaknesses:**

1. Although the proposed method demonstrates strong performance, all the compared baselines are relatively outdated (V2VNet, F-Cooper, V2X-ViT, etc.), which weakens the persuasiveness of the paper.

2. The authors did not provide a clear table to present the experimental results. Providing a comprehensive table comparing the proposed method with the baselines would better clarify the experimental results

**Questions:**

1. Why didn’t the authors compare their method with some recent approaches from 2025, such as STAMP [1], PolyInter [2], and others?

2. Although the proposed method is claimed to handle heterogeneous features, the experiments only involve LiDAR as the sensor. I am curious whether it can effectively process other sensors, such as RGB cameras. The authors do not need to provide comparative experiments. And verification would be enough.

3. The authors should provide more detailed training parameters to ensure the reproducibility of the method, such as the learning rate, optimizer, and others.


[1] Xia Y, Yuan Q, Luo G, et al. One is Plenty: A Polymorphic Feature Interpreter for Immutable Heterogeneous Collaborative Perception[C]//Proceedings of the Computer Vision and Pattern Recognition Conference. 2025: 1592-1601.

[2] Gao X, Xu R, Li J, et al. Stamp: Scalable task and model-agnostic collaborative perception[J]. arXiv preprint arXiv:2501.18616, 2025.

---

### Note · Authors · 2025-11-12

I have read and agree with the venue's withdrawal policy on behalf of myself and my co-authors.